# Genomic Analysis of Vavilov’s Historic Chickpea Landraces Reveals Footprints of Environmental and Human Selection

**DOI:** 10.3390/ijms21113952

**Published:** 2020-05-31

**Authors:** Alena Sokolkova, Sergey V. Bulyntsev, Peter L. Chang, Noelia Carrasquilla-Garcia, Anna A. Igolkina, Nina V. Noujdina, Eric von Wettberg, Margarita A. Vishnyakova, Douglas R. Cook, Sergey V. Nuzhdin, Maria G. Samsonova

**Affiliations:** 1Department of Applied Mathematics, Peter the Great St. Petersburg Polytechnic University, 195251 St. Petersburg, Russia; alyonasok@yandex.ru (A.S.); igolkinaanna11@gmail.com (A.A.I.); nnoujdina@gmail.com (N.V.N.); 2Federal Research Centre All-Russian N.I. Vavilov Institute of Plant Genetic Resources (VIR), 190000 St. Petersburg, Russia; s_bulyntsev@mail.ru (S.V.B.); m.vishnyakova@vir.nw.ru (M.A.V.); 3Dornsife College of Letters Arts & Sciences, Program in Molecular and Computational Biology, University of Southern California, Los Angeles, CA 90089, USA; peterc@usc.edu; 4Department of Plant Pathology, University of California Davis, Davis, CA 95616, USA; noecarras@ucdavis.edu; 5Department of Geography, University of California Los Angeles, Los Angeles, CA 90095, USA; 6Department of Plant and Soil Science, University of Vermont, Burlington, VT 05405, USA; Eric.Bishop-Von-Wettberg@uvm.edu

**Keywords:** bioclimatic analysis, chickpea, GBS, GWAS, haploblock, SNP

## Abstract

A defining challenge of the 21st century is meeting the nutritional demands of the growing human population, under a scenario of limited land and water resources and under the specter of climate change. The Vavilov seed bank contains numerous landraces collected nearly a hundred years ago, and thus may contain ‘genetic gems’ with the potential to enhance modern breeding efforts. Here, we analyze 407 landraces, sampled from major historic centers of chickpea cultivation and secondary diversification. Genome-Wide Association Studies (GWAS) conducted on both phenotypic traits and bioclimatic variables at landraces sampling sites as extended phenotypes resulted in 84 GWAS hits associated to various regions. The novel haploblock-based test identified haploblocks enriched for single nucleotide polymorphisms (SNPs) associated with phenotypes and bioclimatic variables. Subsequent bi-clustering of traits sharing enriched haploblocks underscored both non-random distribution of SNPs among several haploblocks and their association with multiple traits. We hypothesize that these clusters of pleiotropic SNPs represent co-adapted genetic complexes to a range of environmental conditions that chickpea experienced during domestication and subsequent geographic radiation. Linking genetic variation to phenotypic data and a wealth of historic information preserved in historic seed banks are the keys for genome-based and environment-informed breeding intensification.

## 1. Introduction

Landraces dominated agriculture for millennia, until the advent of intensive modern breeding in the mid 20th century, when reduced sets of elite cultivated varieties largely displaced the wider diversity of local genotypes [1]. Although the shift away from landraces was neither systematic nor synchronous, it is generally accepted that the subsequent convergence on a limited set of elite germplasm removed considerable useful variation [2]. In the early 20th century (1911–1940), N.I. Vavilov led a systematic effort to collect and preserve crop diversity, now maintained within the Vavilov Institute of Plant Genetic Resources (VIR) collection in St. Petersburg, Russia [3]. The geographic distribution and genetic diversity of most crops collected during this time frame are likely to reflect their historic patterns of cultivation established over the preceding millennia. Exploring these unique genetic resources provides an opportunity to revisit hypotheses about the radiation and secondary diversification of crops, not possible using later collections. Moreover, the expanded diversity of these early collections likely contains ‘genetic gems’ with the potential to enhance modern breeding efforts [4].

Here, we focus on biodiversity of *Cicer arietinum*, chickpea, which is among the world’s most widely grown grain legumes and provides a vital source of dietary protein for ~15% of the world’s population. Chickpea was first domesticated ~10 KYA, initially in southeastern Turkey, and then spread regionally throughout the Fertile Crescent. Although exact dates are unknown, archeological evidence suggests chickpea moved to India ~6000 years ago and to Ethiopia and North Africa ~3000 years ago [5]. Millennia of cultivation in these new areas, largely in isolation from each other, led to the establishment of new centers of secondary diversity, with accompanying differentiation of regionally specific landraces. Despite this generally accepted scenario, the relationships among the chickpea crops at these historic centers of cultivation are not fully resolved.

Chickpea domestication and breeding imposed a severe genetic bottleneck on the crop, with an estimated >95% of diversity lost between the crop wild progenitor and modern elite varieties [6]. Landraces represent an intermediate step to modern germplasm. An implicit, yet untested assumption is that chickpea landraces will have increased genetic diversity relative to modern elite germplasm. Moreover, we posit that geographic patterns of landrace diversity were shaped by post-domestication selection to adapt the crop to different agro-ecological environments and cultural preferences. Although Vavilov was unable to quantify the extent of diversity and differentiation, he and his contemporaries recognized the value of landraces as reserves of agriculturally-relevant traits, which motivated these early efforts in collection and conservation. Thus, chickpea landraces are expected to contain beneficial alleles, not segregating among modern elite varieties, which can be accessed and prioritized for crop improvement using genomics, phenotyping, and computational methods.

Here, we combine genomics, phenotyping, and computational biology to understand chickpea’s agricultural variation one century ago, and from that analysis to infer the breadth and genetic bases of trait variation in the pre-modern era. Such knowledge can prioritize landrace haplotypes that contributed to diversification of chickpea as a crop, particularly haplotypes missing from modern breeding programs, thereby facilitating their use for crop improvement.

## 2. Results

### 2.1. Germplasm Resources and Phenotyping

To fully cover the biogeographic range of historic chickpea cultivation, we assembled 407 accessions collected between 1911 and 1940. Text descriptions of sampling locations, which were often local markets in small towns, were converted to geographic coordinates (Figure 1a). This set of accessions is enriched for genotypes under cultivation a minimum of one century ago in Turkey, India, Ethiopia, Uzbekistan, and Morocco, representing the major centers of post-domestication chickpea diversification and comprising 55% of the 407 analyzed accessions. Beyond the 147 Turkish and Ethiopian genotypes analyzed in an earlier study [4], we genotyped and/or phenotyped an additional 260 accessions spanning a total of 30 countries, with adjacent countries occasionally representing single extended historic agricultural systems (for examples, Ethiopia and Eritrea in eastern Africa, and several countries from the Fertile Crescent) (Appendix A). The entire set of accessions was phenotyped under field conditions, genotyped, and used for further analysis.

Correlation analyses of nineteen bioclimatic variables (bioclimatic variables and their abbreviations are presented in Appendix A) from the range of chickpea collection sites revealed five groups of correlated variables (Figure 1b; Appendix A). Three bioclimatic variables (BIO_2_, BIO_19_, DEM) were not strongly correlated to other variables. The first, third, and fifth groups (Appendix A) correspond to temperature traits. The second and fourth groups (Appendix A) consist of precipitation variables. While the first group (Appendix A) consists of traits with moderate positive correlation (pairwise Spearman correlation coefficient, r > 0.4, Figure 1b), traits in the second group (Appendix A) have stronger positive correlations (pairwise Spearman correlation coefficient, r > 0.7, Figure 1b), and traits in the remaining groups (Appendix A) have the strongest positive correlations (pairwise Spearman correlation coefficient, r > 0.9, Figure 1b).

All 407 landraces accessions were phenotyped for thirty-six traits under field conditions in Kuban, Russia. The scored phenotypes and their abbreviations are presented in Appendix A. Correlation analyses identified three groups of correlated traits (Figure 2). Phenotypic traits related to the color of plant organs and tissues were moderately correlated (pairwise Spearman correlation coefficient, r > 0.5, Figure 2) and form a single group. Quantitative traits characterizing the weights and sizes of whole plants and pods, as well as leaf size, also had moderate positive correlations (pairwise Spearman correlation coefficient, r > 0.4, Figure 2) and form two groups. Two phenological traits describing the duration of flowering and the duration of pod maturation had strong negative correlation (Spearman correlation coefficient, r = −0.76, Figure 2). Pod shape (PodSH) had moderate negative correlation with pod length (PDL) (Spearman correlation coefficient, r = −0.53, Figure 2) and pod width (PDW) (Spearman correlation coefficient, r = −0.55, Figure 2). Pod shape also had moderate negative correlation with thousand seeds weight (TSW) (Spearman correlation coefficient, r = −0.47, Figure 2). Phenotypic traits related to organ and tissue coloration had moderate negative correlation with traits describing the weights and sizes of plant and pods (pairwise Spearman correlation coefficient, r < −0.4, Figure 2).

### 2.2. Marker Polymorphism Analysis

Restriction site associated genotyping by sequencing (RAD-GBS) was used to survey polymorphism within the genomes of 407 accessions. SNPs were filtered to retain polymorphisms present in at least 90% of genotypes with a minor allele frequency of at least 3%. The resulting 2579 polymorphisms are distributed among all chromosomes, but with variable density that is especially elevated on chromosome 4 (Figure 3a). The elevated polymorphism content of chickpea chromosome 4 has been observed in previous studies (e.g., [4]). We hypothesized that selection and introgression via inadvertent hybridization between more and less advanced morphotypes might have resulted in agricultural improvement genes being aggregated to genomic ‘agro islands’, and in genotype-to-phenotype relationships resembling widespread pleiotropy.

The sufficiency of this marker set for genetic tests depends in part on the scale of linkage disequilibrium (LD), because the relationship between physical distance and recombination frequency determines the precision of genetic association tests. LD is the non-random association between polymorphisms and can originate from demographic processes (e.g., shared ancestry and drift) or from selection (i.e., selective sweeps). In smaller populations of predominantly selfing organisms (including those that are the product of breeding), drift and selection typically have stronger effects than recombination, and thus LD extends to large genomic regions. Landraces are expected to exhibit especially extended LD. In line with these expectations, LD in chickpea landraces is very slow to decay (Figure 3b; Appendix A). Moreover, the marker density is uneven between chromosomes: from 91 SNPs on chromosome Ca8 to 792 SNPs on chromosome Ca4 (Figure 3c). Our sample size is comparable with other recent GWAS crop publications, hopefully resulting in adequate power.

### 2.3. Geographic Analyses

Patterns of population differentiation were analyzed using principle components (PCA) and visualized with unrooted trees. Figure 4 depicts the PCA plot for genetic data of the first versus second components and Appendix A depicts a summary of variation and covariation attributed to the first five principle components. Interestingly, the accessions from the center of domestication, Turkey, are mainly divided into two clusters with light seeded Kabuli and Desi, which are smaller with dark seeds and purple flowers market classes intermixed with each cluster (Figure 4). The lack of distinctiveness between Desi and Kabuli adds further support to the same conclusion reached by Penmetsa et al. [7]. All groups containing Turkish accessions also contain minor representation from other regions, with the exception of a preponderance of landraces from North Africa in one of the Turkish groups. Notably, landraces from India and Ethiopia, which represent two of Vavilov’s major sites of secondary diversification [8], are well resolved, though not exclusive of one another. Turkish accessions are absent from the group of Ethiopian landraces and constitute only a minor component of the Indian group, which is instead enriched in landraces from Central Asia. A portion of Central Asian accessions also occur in a distinct grouping dominated by the ancestral Desi form (Figure 4).

These observations are consistent with the deduced pattern of molecular evolution. Maximum likelihood phylogenetic trees constructed with genome-wide SNP (Figure 5a) support inferences from the PCA analysis. Central Asian and Turkish accessions are broadly distributed throughout the tree, but notably absent from groups predominated by India and Ethiopia, consistent with more extensive diversity (Appendix A) at the Turkish center of origin for the species, and with longstanding, but distinct secondary diversification in India, Central Asia, and Ethiopia. Chromosome 4 is known to have excess diversity relative to the rest of the genome [9,10], as indeed we observe here. Interestingly, certain of the relationships observed using genome-wide SNP are obscured in the tree constructed from chromosome 4 SNPs (Figure 5b). In particular, the previously coherent group of Ethiopian genotypes is divided more broadly within the tree and there is both greater subdivision within the Indian group and less distinction from the Central Asian landraces.

### 2.4. Single Trait Associations

Genetic and phenotypic data were strongly concordant, as described in Appendix A, which shows co-variances between genetic and phenotypic data.

To account for these effects, GWAS analysis was implemented with the first eight PCA axes scores used as covariates for all phenotypic and bioclimatic data (Appendix A), revealing multiple significant associations among 70 SNPs with bioclimatic and phenotypic traits (Figure 6 and Figure 7; Appendix A). Twelve of 70 markers were found to have significant associations with two or more traits. SNP Ca2: 17161867 is associated with plant weight without pods (WpWp) as well as isothermality (BIO_3_) and mean temperature of the warmest quarter (BIO_10_) (see Appendix A for a full list of bioclimatic variables and phenotypes abbreviations). These genetic findings are supported by WpWp weakly negatively correlated with BIO_3_ and BIO_10_. SNP Ca3: 20549509 and SNP Ca6: 2908823 are associated with mean diurnal range (BIO_2_) and BIO_3_, which are themselves weakly positively correlated (Figure 1b). Three SNPs, two on the 8th chromosome (SNP Ca8: 9098790 and Ca8: 10314452) and one on the 4th chromosome (SNP Ca4: 30948593), are associated with two phenotypic variables: biological yield (Byld) and plant weight without pods (WpWp), which are very strongly correlated and appear to derive from common genetic capacities (r = 0.92; Figure 2). Also on chromosome 4, Ca4: 33967674 is associated with the correlated group of phenotypes that includes plant weight traits (weight of seeds, pods, and the whole plant). SNP Ca6: 57117312 is associated with flower color (FloCol) and seed shape (SSH), which are themselves moderate negatively correlated (r = −0.45, Figure 2). SNP Ca7: 30930779 is associated with BIO_3_, number of seeds per plant (SPP), and the group of phenotypes characterizing plant and organ weights. Three additional SNPs on chromosome 7 (SNP Ca7: 33337524, Ca7: 33340372, Ca7: 33457287) are associated with three bioclimatic variables, BIO_3_, BIO_6_, and BIO_11_, which are part of a larger group of correlated variables (Figure 1b).

To incorporate geography explicitly into the analysis, we repeated the above GWAS, but with the addition of the first two axes of PCoA, which derive from the analysis of landrace geographic variation (Appendix A). The results of these analyses were generally consistent with the results described above and are only introduced briefly here. An additional set of significant associations was found. Twelve SNPs are associated with pod length (PDL), nine on chromosome 6 and three on chromosome 7. Ten of these twelve SNPs exhibit significant linkage. Two SNPs on chromosome 7 are associated with secondary branching (StemBranch2order), but without strong linkage.

Because of extended LD, we cannot identify causal relationships between SNPs and phenotypes. Nevertheless, we explored the potential nature of the associated genes and found several important genes that have been reported in previous studies. For example, genes Ca_10410, Ca_10426, and Ca_10428 are present within haploblock Ca6:2541669….Ca6:3024335, to which several SNPs associated with the beginning to flowering to the beginning to maturation phenotype and temperature related variables map (see Appendix A). Ca_10410 (Ca6:2766285….2768999) is involved in floral development and encodes flavin-binding kelch repeat F-box protein with high homology to circadian clock-associated FKF1 gene of soybean. Ca_10426 (Ca6:2881369….2884463) encodes a XAP5 protein important for light regulation of the circadian clock that plays a global role in coordinating growth in response to the light environment. SNP Ca2: 17161867 associated with plant weight without pods (WpWp) and temperature related bioclimatic variables BIO_3_ and BIO_10_, as well as Ca2: 17161884 associated with the duration of flowering (BegFloEndFlo) and BIO_3_ are all located within intron of gene Ca_16015. This gene encodes phosphoenolpyruvate carboxylase, enzyme involved in carbon fixation, and citric acid cycle biosynthesis flux [11]. The first intron of Ca_11533 gene encoding beta-D-xylosidase contains SNP Ca8: 9098790, which is associated with both WpWp and Byld. beta-D-Xylosidases are involved in the breakdown of xylan, a major component of plant cell-wall hemicelluloses [12]. SNP Ca1: 2218700, which is associated with WpWp, is located in the intergenic region upstream of gene Ca_00278 that encodes protein with polyphenol oxidase activity. In *Clematis terniflora* DC, decreasing activity of this enzyme elevates the plant photosynthesis by activating the glycolysis process, regulating Calvin cycle, and providing adenosine triphosphate (ATP) for energy metabolism. Besides, polyphenol oxidase is involved in the formation of brown melanin pigment in fruits and vegetables, plays a crucial role in the biosynthesis of secondary metabolites, and has a role in plant defense against biotic and abiotic stresses [13]. SNP Ca3: 10855323 associated with WpWp is located upstream of Ca_19358 gene encoding beta-*N*-acetylhexoamidase that catalyzes the hydrolysis of *N*-acetylglucosamine or *N*-acetylgalactosamine from the non-reducing terminal of oligosaccharides, glycoproteins, glycolipids, and other glycoconjugates. b-*N*-acetylhexosaminidase is highly active in dry or germinating seeds, where it participates in the degradation of reserve glycoproteins. Moreover, its activity is induced in the period of ripening in tomato and peaches [14]. The Ca_11539 (Ca8:9151680.…9159194) intron contains several SNPs associated with WpWp. This gene encodes an oligopeptidase degrading short peptides. SNP Ca4: 2145082 associated with flower color (FloCol) is located upstream of Ca_07836 gene, which is homologue of genes in *Pisum sativum* (protein A) and *Medicago truncatula* (bHLH-A), which are flower color associated genes [15].

### 2.5. Clustering of Phenotypes and Variables Sharing Enriched Haploblocks

The total number of the Haploview-inferred [16] haploblocks was 224, encompassing 1264 SNPs (mean per haploblock = 5.6) (Appendix A). Filtering for more than six SNPs left 74 haploblocks (33% of total) as input to find haploblocks enriched for associated SNPs for each trait and variable using the fast gene set enrichment (FGSEA) method [17] (parameter for permutations = 100,000) (Appendix A). Subsequent to bi-clustering of phenotypes and variables sharing enriched haploblocks, we defined several visually distinguished groups (Figure 8, Appendix A). The first group contained two consecutive reproductive stages of plant development: the duration of flowering (BegFloEndFlo) and the duration from the end of flowering until the beginning of maturation (EndFloBegMatu). We hypothesize that the same genetic mechanisms influence the duration of both stages. The second group contains pod shattering (PodShat) and pod drop (PodDrop) traits as well as one-third of all bioclimatic factors, related to both temperature and precipitation, exclusive to a well correlated set from Figure 1b (BIO_6,8,11,12,13,16_). Pod-related traits form a subgroup with three temperature-related bioclimatic factors: mean temperature (BIO_1_), mean temperature of coldest month (BIO_6_), and temperature annual range (BIO_7_); this subgroup is similar in a set of enriched haploblocks with the group containing two additional heat-related bioclimatic factors, max temperature of warmest month (BIO_5_), and mean temperature of warmest quarter (BIO_10_). This grouping is consistent with a well-known relationship between high temperature and pod shattering/retention. A third group includes color-related traits, flower color (FloCol), peduncle color (FlowstemColo), seed color (SCO), and stem color (StemColo), which is expected, because genes in the phenylpropanoid pathway are implicated in the production of pigments in different plant organs. A fourth group aggregates *Ascochyta* blight resistance (AscoRes) and precipitation of the coldest quarter (BIO_19_), which reflects a well understood relationship between *Ascochyta* incidence and rainfall during periods of reduced temperatures. Also of note is a group containing moisture stress-related covariates (BIO_14,17_, precipitation of the driest month/quarter) and plant height (Ptht), which is expected to depend on moisture availability; interestingly, this group clusters with a group that contains phenotypic traits related to plant size (biological yield and pod size), which are traits related to the duration of vegetative growth and that are limited by moisture availability.

## 3. Discussion

For many millennia, farmers and breeders have focused on selecting crops with desirable phenotypes [2]. With the successful domestication of numerous crops came the incremental loss of genetic and phenotypic variation. Genetic bottlenecks are especially common in selfing species such as grain legumes (e.g., [18]). Novel sources of variation for biotic and abiotic stress resistance are especially needed in chickpea, because the crop is often grown by resource-poor farmers, on marginal lands, and under low-input conditions. Broadening chickpea’s genetic base should facilitate production of new varieties to address these needs, while also meeting changing consumer demands, new agricultural practices, and anticipated shifts in climatic conditions [6].

Chickpea landraces represent an expanded source of genetic and phenotypic variation that has not been systematically explored and has been used only in an ad hoc manner for modern breeding. The Vavilov Institute of Plant Genetic Resources is one of the world’s primary libraries of lost genetic variation in food crops, capturing the genetic and functional diversity of regionally stratified agriculture typical of one century ago. It contains tens of thousands of legume accessions, including approximately one thousand chickpea accessions collected prior to intensive international breeding efforts [3]. The re-introduction of genetic material from the Vavilov Institute’s collection into modern elite varieties could be a potent force for future agricultural improvement. To this end, we combine genomics, phenotyping, and computational biology to characterize the chickpea collection of Nikolay Vavilov and his colleagues, linking traits and environments to genes. Our results highlight the collection’s currently latent potential of chickpea landraces, and underscore the value of this resource to meet the enormous challenges of 21st century agriculture. However, the identified candidate genes are needed in further validation and functional confirmation owing to such factors as one-year observation of phenotypes and long extend of LD in the germplasm.

Our observations contribute to an increasing understanding of genetic variation of quantitative and categorical traits in chickpea [19,20,21]. The present work adds a new dimension by incorporating a wider set of historical crop diversity, and by treating bioclimatic data at accession sampling sites as extended crop traits. In doing so, our GWAS hits highlight associations to genomic regions not discovered in prior GWAS and quantitative trait locus (QTL) analyses (Appendix A). These hits map in the vicinity of genes involved in floral development, photosynthesis, cell wall or secondary metabolism, and carbohydrate biosynthesis, and some of them are close to already known QTLs. For example, SNP Ca4: 33967674, associated with yield, pod weight, plant weight without pods, and seed weight per plant, is located 752 kb downstream from known QTL (Appendix A) governing pod number trait [22] and SNP Ca3: 28094292, associated with plant weight without pods, localizes 96 kb downstream of QTL (Appendix A) containing cluster of FLOWERING LOCUS T (FT) genes and controlling phenology and growth habit [23]. SNP Ca4: 30948593 and SNP Ca8: 10314452, associated with yield, are located ~90 kb upstream from previously detected SNP (Appendix A) and ~25 kb downstream from previously detected SNP, respectively (Appendix A), also associated with yield [24]. SNP Ca6: 3024192, associated with beginning of flowering to the beginning of maturation phenotype, is located in the same haploblock Ca6_Block_3 (~87 kb upstream) as the previously detected SNP (Appendix A), associated with days to 50% flowering [24]. Previously, we [25] published a study in which we were looking for associations between SNPs and bioclimatic covariates at collection sites. Two covariates, which include temperature characteristics, were jointly associated with one SNP on chromosome 8 (Ca8: 10314452). This SNP is associated with two phenotypic variables: biological yield (Byld) and plant weight without pods (WpWp) in the current study.

To rigorously test for associations, we implement a novel haploblock-based test that, we believe, will find much use in the crop genomics. The underlying statistics for the test are similar to the gene set enrichment analysis, where each haploblock represents a set of SNPs associated with a trait and all SNPs are ranked according to GWAS *p*-values. This analysis identified eleven haploblocks (Appendix A) intersecting with previously reported GWAS hits. Haploblock Ca1_Block_18 and haploblock Ca4_Block_18 are enriched for SNPs associated with several phenotypes and bioclimatic variables, including thousand seeds weight phenotype. These haploblocks covers SNP on chromosome 1 and SNPs on chromosome 4, respectively, reported by Varshney et al. [24], associated with 100 seed weight (Appendix A). Haploblock Ca3_Block_4, haploblock Ca4_Block_54 and haploblock Ca5_Block_4 are enriched for SNPs associated with several phenotypes and bioclimatic variables, including seeds weight per plant phenotype. These haploblocks overlay four SNPs on chromosome 3, three SNPs on chromosome 4, and eight SNPs on chromosome 5, respectively, reported by Varshney et al. [24], associated with yield per plant (Appendix A). Haploblock Ca3_Block_7 is enriched for SNPs associated with the duration of vegetative growth, with seeds weight per plant, and with three bioclimatic variables (BIO_5_, BIO_13_, BIO_16_). This haploblock covers two SNPs on chromosome 3, reported by Varshney et al. [24], associated with days to 50% flowering and with yield per plant, respectively (Appendix A). Haploblock Ca3_Block_16 is enriched for SNPs associated with the duration of vegetative growth, as well as with plant height, plant weight without pods, and temperature-related bioclimatic variables BIO_3_ and BIO_5_. This haploblock intersects with a QTL for days to 50% flowering time (Appendix A) reported from the GWAS analysis of Upadhyaya and colleagues [19]; Upadhyaya et al. nominated a particular candidate gene, SBP (SQUAMOSA promoter binding protein), though we advocate a more cautious approach that recognizes limitations of the study design and instead implicates haplotype intervals. Haploblock Ca4_Block_9 is enriched for SNPs associated with the duration of vegetative growth, with pod shattering, and with four bioclimatic variables (BIO_4_, BIO_6_, BIO_7_, BIO_12_). This haploblock covers SNP on chromosome 4 associated with days to 50% flowering (Appendix A), reported by Varshney et al. [24]. Haploblock Ca7_Block_12 is enriched for SNPs associated with the duration of vegetative growth, with number of seeds per plant, with stem branchness, and with temperature-related bioclimatic variable BIO_3_. This haploblock covers SNP on chromosome 7 associated with days to maturity (Appendix A), reported by Varshney et al. [24]. The last haploblock, Ca8_Block_7, is enriched for traits related for branching and covers SNP on chromosome 8 reported by Bajaj et al. [20], associated with branch number (Appendix A).

Previously, we [4] published a pilot study combining historic phenotypic data with reduced representation sequencing to establish a proof-of-principle for the results reported here. We employed a combination of genomics, computational biology, and phenotyping to characterize VIR’s 147 chickpea accessions from Turkey and Ethiopia, representing chickpea’s center of origin and a major location of secondary diversification, respectively. The majority of SNPs associated with multiple traits localized to a single chromosome 4 region. Here, we observe similar patterns with a larger sample of more diverse landraces and with a more comprehensive phenotypic and environmental dataset. We find multiple SNPs that are non-randomly distributed among several haploblocks, many of which are associated with multiple phenotypes (Appendix A). The non-random clustering of phenotypes and variables (Figure 8) exactly arises as a result of such multi-trait associations. Although the grouping of traits and ancestral bioclimatic variables does not necessarily imply co-selection during domestication (e.g., [26]), these clusters may represent genetic complexes co-adapted to a range of environmental conditions that chickpea experienced during domestication and subsequent geographic radiation. Indeed, many of the trait–environment associations reflect well-known interactions between environmental factors and the crop’s biology; for example, the relationships between *Ascochyta* blight occurrence and the duration of cool-wet periods, as well as the increased incidence of pod abortion and shattering under conditions of heat stress. Thus, by combining genomics with an explicit biogeographic framework encompassing climatic and phenotype covariates, we are able to suggest concordance between human selection, the crop’s known biology, and environmental constraints.

## 4. Materials and Methods

### 4.1. Germplasm Resources and Phenotyping

We assembled a collection of VIR’s chickpea germplasm originating from a range of countries including Ethiopia, Lebanon, Morocco, Turkey, India, and the broader Central Asia and Mediterranean regions (see Appendix A). Phenotyping of the 407 chickpea genotype collection was conducted at the VIR Kuban experimental station with climatic conditions well suited for chickpea cultivation (see Text S1). During the vegetative period, thirty-six phenological, morphological, agronomical, and biological descriptors were measured. The scored phenotypes and their abbreviations are presented in Appendix A.

### 4.2. Genotyping by Sequencing (GBS) and SNP Calling

The restriction site associated (RAD) GBS protocol from von Wettberg et al. [6] was used to generate reduced representation sequence data for 407 accessions (see Text S2). All Illumina data are available from the National Center for Biotechnology database under BioProject PRJNA388691. SNPs were called using the Genome Analysis Tool Kit (GATK) pipeline [27] and further filtered with VCFtools [28]. A total of 2579 SNPs accessions passed all filters, with 407 accessions remaining for further analysis.

### 4.3. Genetic Data Analyses

Principal component analysis (PCA) was conducted using the “SNPRelate” R library [29]. Custom scripts in Python [30] and R [31] were used to plot depth and distribution of SNPs on chromosomes.

Linkage disequilibrium (LD) was estimated using the squared correlation coefficient (r^2^) between genotypes. VCFtools [28] was used to calculate intra-chromosomal and unlinked r^2^ values. LD decay was assessed by plotting intra-chromosomal r^2^ values against the physical distance (bp) between markers. The parametric 95th percentile of unlinked r^2^ values distribution was taken as a critical value. The threshold beyond which the LD was accepted as real physical linkage was estimated to be r^2^ = 0.16. The intersection of the smothering second degree local regression (LOESS) curve of intra-chromosomal r^2^ values with this threshold was considered to be an estimate of the range of LD. 

Relationships among genotypes were calculated and maximum likelihood phylogenetic trees were constructed using SNPhylo [32] based on filtered SNPs and drawn using R libraries “phytools” [33] and “ape” [34].

The nucleotide diversity (pi) was estimated from polymorphic sites and separately for each chromosome and geographical group using VCFtools [28]. By considering only polymorphic sites, we overestimate genomic diversity; however, these estimations can be used for between group comparisons. We applied the Mann–Whitney–Wilcoxon test [35] to make between group comparisons.

The Genome-wide complex trait analysis (GCTA) program [36] was used to estimate the proportion of variance in phenotypes explained by all genome-wide SNPs. First, phenotypic data were normalized. Then, the genetic relationships among individuals from genome-wide SNPs were calculated using GCTA-GRM (genetic relationship matrix) analysis. Finally, GCTA-GREML (genome-based restricted maximum likelihood) analysis was performed to estimate the proportion of variance in a phenotype explained by all GWAS SNPs (i.e., the SNP-based heritability). 

### 4.4. Bioclimatic Analysis

Bioclimatic analysis was performed as described in Plekhanova et al. [4]; for details, see Text S3. Nineteen quantitative bioclimatic variables were used in the analysis (Appendix A).

Shapiro–Wilk test for normality [37] was implemented to quantitative phenotypic traits and quantitative bioclimatic variables. Spearman correlation coefficients were calculated using the “rcorr” function from the “Hmisc” R library [38].

### 4.5. Mapping Approaches

GWAS analysis was performed using a single-locus linear mixed model, implemented in FaST-LMM toolset (factored spectrally transformed linear mixed models) [39]. Principal component analysis (PCA) of 2579 SNPs revealed that the first eight significant principal components (PCs) explained 48% of the variance of all markers. The LMM model was implemented with the first eight PCA axes scores used as covariates for all phenotypic and bioclimatic data. Principal coordinate analysis (PCoA), based on geographical distances between the accessions, was performed using the “pco” function from the “labdsv” library [40] in R, and revealed that the first two significant PCs explained 59% of the variance. We repeated the GWAS analysis including the first eight PCA axes scores and the first two PCoA axes scores as covariates for all traits. In both cases, we used genomic control parameter (λ_GC_) and a false discovery rate (FDR) [41] of 0.05 to determine significant trait associated loci separately for each trait. Manhattan plots were performed using “CMplot” library [42] in R.

Annotation of significant associated markers was performed using the SNPEff program [43], as well as the legume information system (LIS) [44] and the LegumeIP [45] databases.

### 4.6. Biogeographic Analyses

In total, 407 accessions were split into six distinct groups reflecting geographic locations (Appendix A): Ethiopia (“ETHI”), India (“IND”), Lebanon (“LEB”), Morocco (“MOR”), Turkey (“TUR”), and Central Asia (“C_ASIA”). The Mann–Whitney–Wilcoxon test [35] was used to identify differences among groups for each bioclimatic variable.

### 4.7. Haploblock Enrichment Analysis and Clustering of Enriched Haploblocks

To divide the genome into haplotype blocks (haploblocks) based on linkage disequilibrium, Haploview tools [16] were applied to the set of 2579 SNPs. Chromosomal regions with strong linkage were identified using default Haploview parameters (confidence interval for LD [0.7, 0.98]). Each haploblock was considered as the set of SNPs located within a given haploblock. We analysed haploblock enrichment for SNPs associated with trait or variable by applying the logic of gene-set enrichment analysis implemented in the FGSEA method [17], which takes as input data the list of all SNPs ranked by increasing GWAS *p*-values and the list of haploblocks. The method returns an enrichment score and FDR corrected *p*-value [41] for each haploblock. We performed FGSEA analysis for each trait (phenotype and bioclimatic variable), and haploblocks significantly enriched for associated SNPs were defined as those having positive enrichment scores and significantly low FDR corrected *p*-values (<0.05). The outcome of this analysis was that each phenotype or bioclimatic variable was characterized by a set of haploblocks significantly enriched with associated SNPs. To obtain groups of phenotypes and variables sharing sets of enriched haploblocks, we applied bi-clustering on the matrix of pairwise similarities between traits. To estimate the degree of overlap between haploblocks enriched for SNPs associated with different traits, we calculated the haploblock simalarity score as a sum of common haploblocks (i.e., haploblocks enriched for SNPs associated with both traits) divided by the sum of all haploblocks significantly enriched for SNPs associated with these two traits.

## 5. Conclusions

The Vavilov seed bank contains numerous landraces collected nearly one hundred years ago, and thus may contain ‘genetic gems’ with the potential to enhance modern breeding efforts. Here, we analyze 407 landraces, sampled from major historic centers of chickpea cultivation and secondary diversification. The collection was grown in the southern European part of Russia in 2016 with climatic conditions well suited for chickpea cultivation. GWAS conducted on both phenotypic traits and bioclimatic variables at landraces sampling sites as extended phenotypes resulted in 84 GWAS hits associated to various regions, most of which were not discovered in prior GWAS and QTL analyses. The novel haploblock-based test identified haploblocks enriched for SNPs associated with phenotypes and bioclimatic variables, of which eleven haploblocks intersect with previously reported GWAS hits on chromosomes Ca1, Ca3, Ca4, Ca5, Ca6, Ca7, and Ca8. Subsequent bi-clustering of traits sharing enriched haploblocks underscored both non-random distribution of SNPs among several haploblocks and their association with multiple traits. We suggest that these clusters of pleiotropic SNPs represent co-adapted genetic complexes to a range of environmental conditions that chickpea experienced during domestication and subsequent geographic radiation. We observed significant genomic diversity in Central Asia, which may have been a bridge for subsequent radiation in India and nearby areas. Linking genetic variation to phenotypic data and a wealth of historic information preserved in historic seed banks are the keys for genome-based and environment-informed breeding intensification.

## Figures and Tables

**Figure 1 ijms-21-03952-f001:**
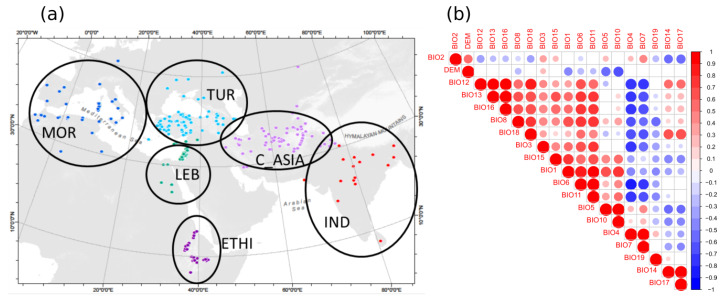
Sample distribution and correlation of bioclimatic variables. (**a**) Location of the chickpea samples around the world that were split into six geographically distinct groups. (**b**) The correlation between nineteen bioclimatic variables (bioclimatic variables and their abbreviations are presented in Appendix A). Color intensity and the size of the asterisk are proportional to the correlation coefficients. ETHI, Ethiopia; IND, India; LEB, Lebanon; MOR, Morocco; TUR, Turkey; C_ASIA, Central Asia.

**Figure 2 ijms-21-03952-f002:**
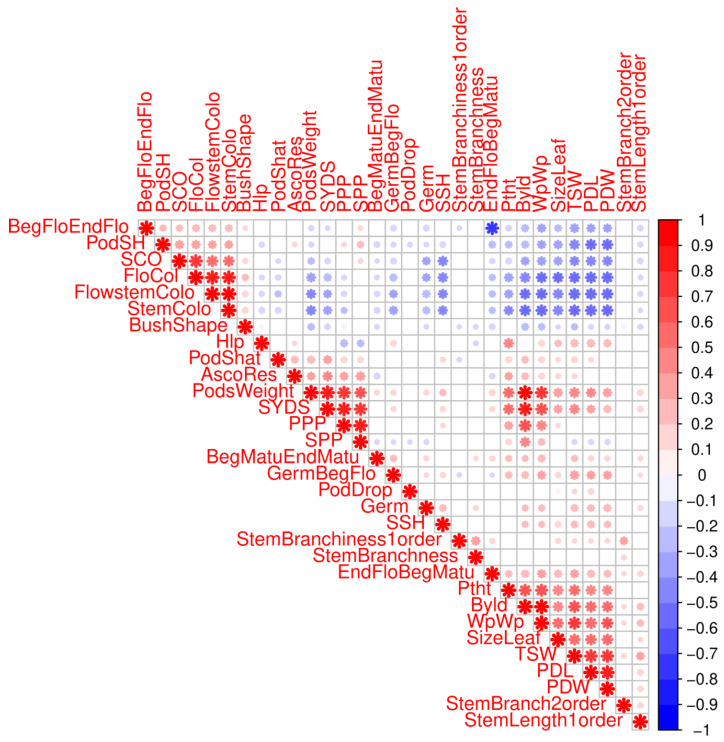
Correlation of thirty-one phenotypic traits. The scored phenotypes and their abbreviations are presented in Appendix A. *Ascochyta*, the degree of damage (AsoDes) trait, was excluded from correlation analysis because it is the opposite value of *Ascochyta* resistance (AscoRes) trait. Moreover, we excluded overlapping time periods traits. Color intensity and the size of the asterisk are proportional to the correlation coefficients. PodSH, pod shape; SCO, seed color; SSP, number of seeds per plant; SSH, seed shape; TSW, thousand seeds weight; PDW, pod width; PDL, pod length.

**Figure 3 ijms-21-03952-f003:**
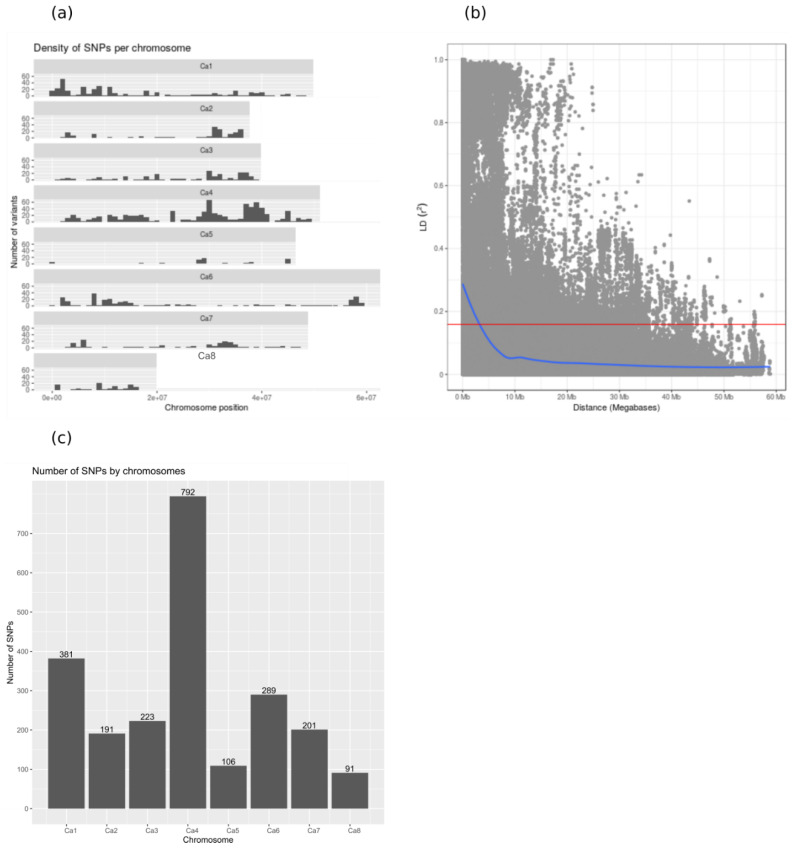
(**a**) Density of SNPs across the chickpea genome. Chromosome Ca6 is the longest chromosome in the chickpea genome (59.46 Mb) and chromosome Ca8 is the shortest (16.48 Mb). (**b**) Linkage disequilibrium (LD) (r^2^) plots of the whole chickpea genome. The horizontal red line indicates the 95th percentile of the distribution of the unlinked r^2^, which gives the critical value of r^2^. (**c**) Distribution of SNPs along the eight chromosomes of the chickpea genome.

**Figure 4 ijms-21-03952-f004:**
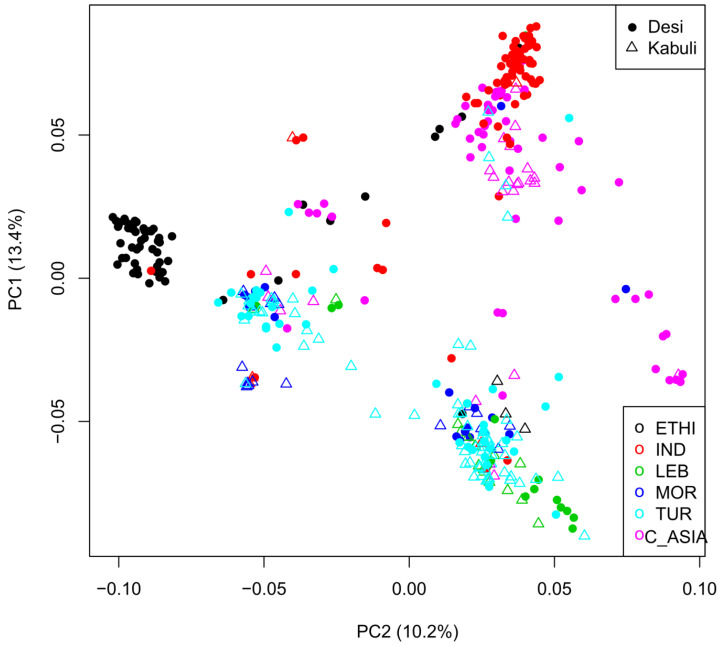
Scatter plots of the first two principal components of the principal component analysis (PCA) based on 2579 SNPs. Each dot represents an accession. Desi varieties are shown as asterisks and Kabuli as triangles.

**Figure 5 ijms-21-03952-f005:**
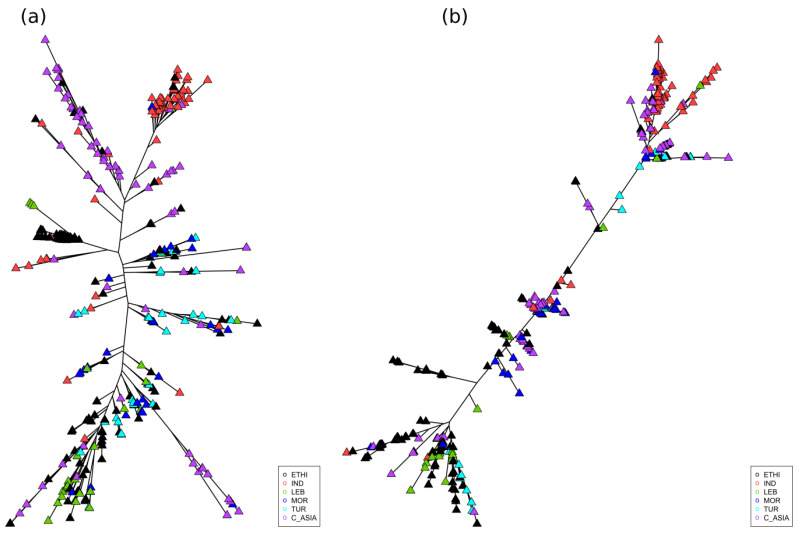
(**a**) Maximum likelihood phylogenetic tree showing relationships among accessions based on the whole genome SNPs and (**b**) on chromosome 4 SNPs.

**Figure 6 ijms-21-03952-f006:**
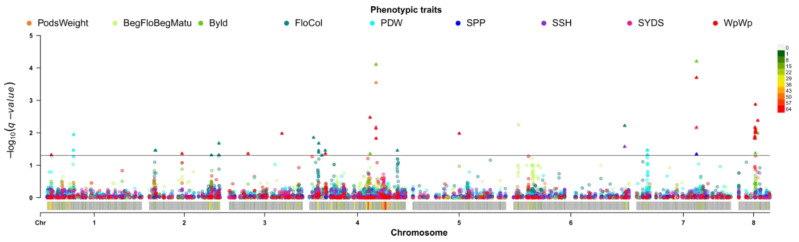
Summary of GWAS analyses with eight PCs as covariates for phenotype data (different colors correspond to different phenotype). SNPs with *q*-value < 0.05 are shown for each chromosome, marked as triangles. Chromosome density is attached on the bottom of the Manhattan plot.

**Figure 7 ijms-21-03952-f007:**
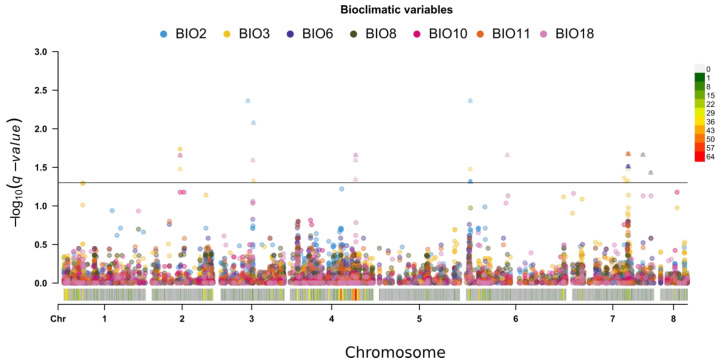
Summary of GWAS analyses with eight PCs as covariates for bioclimatic variables (different colors correspond to different bioclimatic variables). SNPs with *q*-value < 0.05 are shown for each chromosome, marked as triangles.

**Figure 8 ijms-21-03952-f008:**
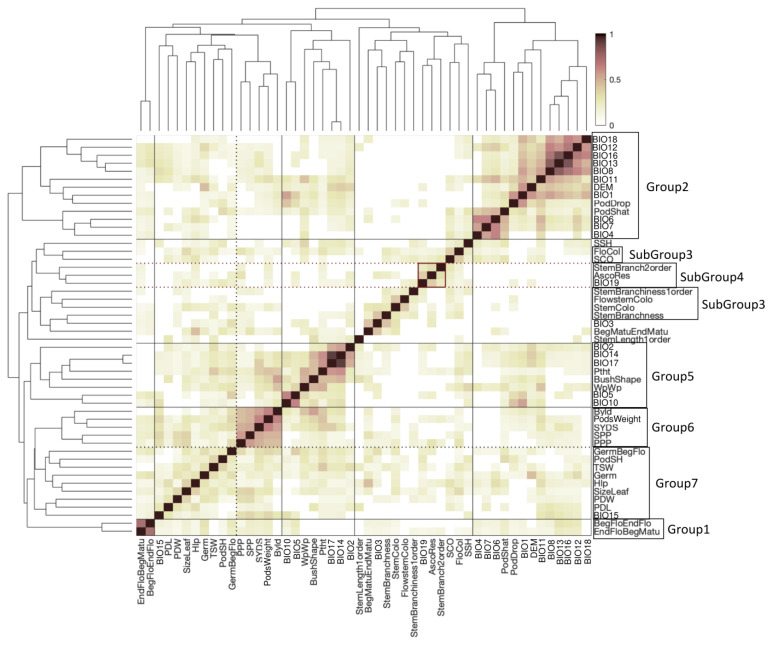
The degree of overlap in haploblocks enriched for SNPs associated with phenotypes and variables. Bi-clustering of similarity scores reveals several visually distinct groups of phenotypes. The haploblock similarity score is defined as a double sum of haploblocks simultaneously enriched for SNPs for both traits normalized to the amount of significantly enriched haploblocks for each trait. The degree of similarity is color coded.

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
