# Peer review of "Genomic Analysis of Vavilov’s Historic Chickpea Landraces Reveals Footprints of Environmental and Human Selection"

_ijms, 2020, doi:10.3390/ijms21113952_

Round 1
Reviewer 1 Report
This paper describes the phenotypic and genetic evaluation of 407 chickpea landraces. The broad diversity of the landraces represented in this study makes it a valuable contribution to the community, and the authors have integrated climatic variables into their GWAS evaluation, which is an interesting approach. They have also presented an approach to identifying haploblocks enriched for SNPs associated with traits/variables. This was followed by a bi-clustering approach to identify groups of traits/variables that shared sets of haploblocks. While the study would obviously have benefitted from a greater number of SNPs, the authors make a reasonable argument that the high LD in domesticated chickpea means that a smaller number of SNPs is still informative. The authors should be congratulated for their work on this interesting set of chickpea accessions.
Overall, this paper has some strong points, and is well-written. However, the bi-clustering approach and the results from this approach need more exposition to be useful to the community. The Discussion section also needs to be revised in some parts to avoid making it into a list of results. I have given specific comments on these and other minor issues below.
Major revisions
Haploblock/Bi-clustering method:
- Section 4.7: At the end of this section, you begin to use the phrase “similarity between traits/variables” and go on to provide a definition based on the sum of common haploblocks, etc. I had to read this section several times to understand that “similarity” had nothing to do with the more simple/common definition of a correlation between traits. I would strongly urge you to find a different phrase to describe what you are getting at here. Perhaps something like “degree of overlap in haploblocks” or whatever it is that you are truly trying to express here. In the legend of Fig. 7 you seem to refer to it as the “haploblock similarity score”, which is more appropriate. The phrase “similarity between traits” is simply too general and it obscures what you are doing in this analysis.
- Pg. 10, L264-290: I have re-read this section several times and spent a good deal of time looking at Figure 7. Unfortunately, I cannot easily find of all the “visually distinguished groups” you discuss here. I can see the grouping for the reproductive stages, and the one for AscoRes and BIO19 (because there is a box around it). However, it was difficult to discern what I should be looking for in the other groups. For example, you mention a group dealing with color, and I spent a good amount of time looking for a tight grouping of FloCol, FlowstemColo, SCO, and StemColo and could not find it. It took me some time to realize that you were referring to a group that was made up primarily of non-color related traits. It’s possible that the heatmap is detracting more from this figure than it is adding. I found it very difficult to trace the small trait labels back up to the dendrograms to see what the groupings actually were. I’m also confused at the dotted lines crossing the figure horizontally that aren’t traced vertically – do these have some significance? This figure needs substantial revision to connect it with the interpretation more directly.
Discussion:
- General comment: The lists of SNP names and haploblock regions in this section detract substantially from the readability here. I understand naming the haploblocks, but I would suggest referring readers to a table rather than listing the coordinates of each block. I would also refrain from listing every single overlapping SNP name and again refer readers to a table.
- Pg. 12, L322-329: This sentence is too long and should be broken up.
- Pg. 12-13, L335-388: This paragraph is a list of each haploblock, the traits/variables that are associated with it, and the SNPs that it overlaps in Varshney et al. Each sentence is well-written, but the paragraph does not seem to carry a broader point other than exhaustively listing these facts one by one. I would suggest shortening this paragraph and focusing on a few of the key haploblocks that you think are the most meaningful findings in this study.
- Pg. 13, L394: A hypothesis must be testable and falsifiable. This is not a hypothesis, but rather a possible explanation of the patterns you see in this study. I’m also not clear on how your reported results speak to hybridization between lines; this concept has not been mentioned previously in the paper.
- Pg. 15, L508: Again, is this a testable, falsifiable hypothesis? If not, please revise.
Minor revisions:
- Results and Fig. 1, Fig. 4, and througout: You sometimes refer to samples as from Uzbekistan and sometimes refer to samples from “Central Asia”. Please choose one of these designations and use it consistently in the text of the paper and in the figure legends.
- Pg. 6, L149-162: The difference between/significance of the Desi vs Kabuli needs more introduction for the non-chickpea expert.
- Fig. 1: Please make the points on the map larger – it’s currently difficult to see what color they are, even when I zoom in on the image.
- Fig. 5: Remove the accession names from these trees and use a branch tip symbol instead to increase clarity.
Author Response
Response to Reviewer 1 Comments
Point 1: Major revisions
Haploblock/Bi-clustering method:
- Section 4.7: At the end of this section, you begin to use the phrase “similarity between traits/variables” and go on to provide a definition based on the sum of common haploblocks, etc. I had to read this section several times to understand that “similarity” had nothing to do with the more simple/common definition of a correlation between traits. I would strongly urge you to find a different phrase to describe what you are getting at here. Perhaps something like “degree of overlap in haploblocks” or whatever it is that you are truly trying to express here. In the legend of Fig. 7 you seem to refer to it as the “haploblock similarity score”, which is more appropriate. The phrase “similarity between traits” is simply too general and it obscures what you are doing in this analysis.
Response 1: We followed the reviewer suggestion and changed this sentence in line 517 as: “To estimate the degree of overlap between haploblocks enriched for SNPs accociated with different traits we calculated the haploblock simalarity score as s a sum of common haploblocks (i.e haploblocks enriched for SNPs associated with both traits) divided by the sum of all haploblocks significantly enriched for SNPs associated with these two traits.”
We also improved the first sentence of the Figure 8 caption (line 310): “The degree of overlap in haploblocks enriched for SNPs associated with phenotypes and variables”. The figure was renamed due to adding new figure to the text.
Point 2: - Pg. 10, L264-290: I have re-read this section several times and spent a good deal of time looking at Figure 7. Unfortunately, I cannot easily find of all the “visually distinguished groups” you discuss here. I can see the grouping for the reproductive stages, and the one for AscoRes and BIO19 (because there is a box around it). However, it was difficult to discern what I should be looking for in the other groups. For example, you mention a group dealing with color, and I spent a good amount of time looking for a tight grouping of FloCol, FlowstemColo, SCO, and StemColo and could not find it. It took me some time to realize that you were referring to a group that was made up primarily of non-color related traits. It’s possible that the heatmap is detracting more from this figure than it is adding. I found it very difficult to trace the small trait labels back up to the dendrograms to see what the groupings actually were. I’m also confused at the dotted lines crossing the figure horizontally that aren’t traced vertically – do these have some significance? This figure needs substantial revision to connect it with the interpretation more directly.
Response 2: We thank the reviewer for the comment and have improved the figure. Dotted lines were made for clarity. We have added the names of groups of phenotypes and variables similar in sets of enriched haploblocks from Supplementary Table S10 to the figure and have changed the name of the figure to Figure 8, because we added a new figure to the text.
Point 3: Discussion:
- General comment: The lists of SNP names and haploblock regions in this section detract substantially from the readability here. I understand naming the haploblocks, but I would suggest referring readers to a table rather than listing the coordinates of each block. I would also refrain from listing every single overlapping SNP name and again refer readers to a table.
Response 3: We thank the reviewer for the comment and have changed the lines 347-355 in Discussion section, in which we deleted the names of overlapping SNPs and the lines 365-412 in which we deleted the coordinates of haploblocks.
Point 4: - Pg. 12, L322-329: This sentence is too long and should be broken up.
Response 4: We thank the reviewer for the comment and split this sentence to two sentences (lines 343-350).
Point 5: - Pg. 12-13, L335-388: This paragraph is a list of each haploblock, the traits/variables that are associated with it, and the SNPs that it overlaps in Varshney et al. Each sentence is well-written, but the paragraph does not seem to carry a broader point other than exhaustively listing these facts one by one. I would suggest shortening this paragraph and focusing on a few of the key haploblocks that you think are the most meaningful findings in this study.
Response 5: We thank the reviewer for the comment and we have shortened this paragraph by deleting the phrase: “Haploblock on chromosome 6, Ca6_Block_3 (2541669….3024335), is enriched for SNPs associated with the duration of seedling growth (beginning of germination – end of germination (Germ) phenotype), the duration of vegetative growth (GermBegFlo phenotype), and with several precipitation related (BIO2, BIO14, BIO17, BIO19) and elevation related (DEM) bioclimatic variables (Table S9). This haploblock overlays Ca6: 29205940 SNP loci regulating diverse plant growth habit (Table S12) reported in the study of Upadhyaya and colleagues [25]” and by deleting the coordinates of haploblocks. Also, we deleted from References paper “25. Upadhyaya HD, Bajaj D, Srivastava R, Daware A, Basu U, Tripathi S, Bharadwaj C, Tyagi AK, Parida SK (2017) Genetic dissection of plant growth habit in chickpea. Funct Integr Genomics. 17(6):711-723”.
Point 6: - Pg. 13, L394: A hypothesis must be testable and falsifiable. This is not a hypothesis, but rather a possible explanation of the patterns you see in this study. I’m also not clear on how your reported results speak to hybridization between lines; this concept has not been mentioned previously in the paper.
Response 6: We removed the sentences about the hypothesis following the reviewer recommendations in lines 419-423 and instead in lines 427-431 provided our explanation why grouping of SNPs associated with phenotypes and bioclimatic variables arise: “The non-random clustering of phenotypes and variables (Fig. 8) exactly arise due to such multi-trait associations. Although the grouping of traits and ancestral bioclimatic variables do not necessarily imply co-selection during domestication (e.g., [26]) these clusters may represent genetic complexes coadapted to a range of environmental conditions that chickpea experienced during domestication and subsequent geographic radiation.”
Point 7: - Pg. 15, L508: Again, is this a testable, falsifiable hypothesis? If not, please revise.
Response 7: We thank the reviewer for the comment and we have changed “hypothesize” to “suggest” in the phrase in line 534 “We hypothesize that these clusters of pleiotropic SNPs represent co-adapted genetic complexes to a range of environmental conditions that chickpea experienced during domestication and subsequent geographic radiation” as we do not aim to check this hypothesis and just provide one of possible explanations of observed phenomenon.
Point 8: Minor revisions:
- Results and Fig. 1, Fig. 4, and througout: You sometimes refer to samples as from Uzbekistan and sometimes refer to samples from “Central Asia”. Please choose one of these designations and use it consistently in the text of the paper and in the figure legends.
Response 8: We thank the reviewer for the comment and have changed designation “Uzbekistan” to “Central Asia” in the text (line 501), in Figures 1, 4 and 5, in Supplementary Table S5 and in Supplementary Figure S1.
Point 9: - Pg. 6, L149-162: The difference between/significance of the Desi vs Kabuli needs more introduction for the non-chickpea expert.
Response 9: We thank the reviewer for the comment and changed the phrase in line 159 to “Interestingly, the accessions from the center of domestication, Turkey, are mainly divided into two clusters with light seeded Kabuli and Desi, which are smaller with dark seeds and purple flowers market classes intermixed with each cluster (Figure 4)”.
Point 10: - Fig. 1: Please make the points on the map larger – it’s currently difficult to see what color they are, even when I zoom in on the image.
Response 10: We thank the reviewer for the comment and have changed the Figure 1a, in which we made splitting of the accessions into six geographically distinct groups more clearly.
Point 11: - Fig. 5: Remove the accession names from these trees and use a branch tip symbol instead to increase clarity.
Response 11: We thank the reviewer for the comment and have changed the Figure 5. We removed the accession names from the trees and used a branch tip symbol instead.
Reviewer 2 Report
The MS conducted a GWAS study for many agronomic and yield-related traits using a relatively large sample size of ~400 chickpea landraces. They have identified many SNP/Haploblocks significantly associated with important traits and discuss potential candidate genes involved. One of the novelties of the MS is they collected and analyzed the relevant bioclimatic variables and included them into the GWAS analysis. I think the MS should be accepted after addressing my comments below.
Major and essential suggestions:
The MS treated the bioclimatic variables as response variables, similar to phenotypic traits in the GWAS analysis. I believe the bioclimatic variables should be treated as covariates in the GWAS model to avoid confounding effect similar to the confounding effect due to population structure. This confounding effect can also be seen in Fig 7 of this MS. Some of the enriched haploblocks of phenotypic traits were similar to the one for bioclimatic variables. For example, the enriched haploblocks for pod drop/shattering might have nothing to do with the underlying genetic of pod drop/shattering; they are just confounding effect coming from BIO18, BIO12,. (Temp., precipitation) etc. They are not relevant for breeding. However, the finding from Fig 7 is useful as it can inform which bioclimatic variables should be included as covariates in the GWAS model. I strongly suggest the authors re-run the GWAS model for pod drop/shattering using BIO18, BIO12, etc. as covariates. This applies to other traits such as AcoRes, ptht, Bushshape, WpWp with a strong correlation with BIOx based on Fig7. The result can then be compared with the previous GWAS. I believe this approach is better to identify the ‘true’ underlying genetic of traits.
I am very surprised that the authors found so many SNP associated with flower color while so few for yield (Figure S3, S4) which contradicts the fact that yield is a complex trait while flower color is a simple Mendelian trait controlled by a few genes (bHLH transcription factor.., Penmetsa RV, 2016 New Phytol) from the phenylpropanoid pathway. It seems that the authors did not point out if any known flower color genes closed to the GWAS hits. If this is not the case, what could be the reason for that?
In the Discussion, the authors dedicated most of the content into the functions of the candidate genes which I think is important. However, I suggest the authors should consider having a small paragraph in the Discussion warning the readers to be cautious about the identified candidate genes and need further validation and functional confirmation due to, no limited to, such as long extend of LD in this germplasm, one-year observation of some complex traits like yield, seed number…
Minor suggestions:
L84, 55% of what? The whole Vavilov collection?
L98 what do you refer to with the first, third and fifth groups
L129 Good to briefly state the reason of elevated SNP in Ca4
L134 What is unlinked r2? Figure 3b is a pool of SNP from the eight chromosomes. It would be good to have a similar figure separate for the eight chromosomes in the Supplementary. This can give a clear picture of LD decay in different chromosomes. Based on the red line, LD decay at around 4Mb which is much longer than other chickpea studies with elite lines (Li et al., 2018 Front.Plnt Sci 9:190). I assume the opposite is true as landrace is supposed to be more diverse and under less selection compared to elite lines. The long extend LD in this MS reduce mapping resolution and give lower confidence for the identified candidate genes.
L146 I am not convinced that these characteristic warrant enough power unless the authors do an appropriate statistical power study
L186 Figure 6b The authors pooled all manhattan plots from different traits into a single plot. This is not very informative. It seems the CMplot package used by the authors can distinguish the traits using different colour or shape.
L186 Figure 6d What does it mean when you find a SNP significantly associated with a bioclimatic variable, let say, precipitation. Remember this variable is a pure environmental condition that doesn’t involve any chickpea or other plants. There is not genetics/physiology involved and is purely statistical association. I think those SNP associations are useless for crop improvement. The bioclimatic variables are actually confounding effect in GWAS and should be included as covariates instead.
L192 The bioclimatic variables are not phenotypic traits, they have no heritability!
L198 The first eight PC explains only about 40% of the total variation (Fig S1). This again points to the author should include bioclimatic variables as covariates as described in major suggestion.
L209 How many replicate in measuring yield and seed number? Be cautious about interpretation based on just one year, one environment.
L236-242 I don’t think this gene has anything to do with flowering and maturation
L267 The physical position of the Haploblocks in Table S9 should be indicated.
L284 There are many QTL detected for Ascochyta blight in chickpea in the literature (a major one in Ca4). Are there any Ascochyta Haploblocks fall in these QTL? There is also no information in term of measuring Ascochyta, how the inoculation was done, what pathotype of the pathogen, ..etc
L442 Is the result reported?
Author Response
Response to Reviewer 2 Comments
Point 1: Major and essential suggestions:
The MS treated the bioclimatic variables as response variables, similar to phenotypic traits in the GWAS analysis. I believe the bioclimatic variables should be treated as covariates in the GWAS model to avoid confounding effect similar to the confounding effect due to population structure. This confounding effect can also be seen in Fig 7 of this MS. Some of the enriched haploblocks of phenotypic traits were similar to the one for bioclimatic variables. For example, the enriched haploblocks for pod drop/shattering might have nothing to do with the underlying genetic of pod drop/shattering; they are just confounding effect coming from BIO18, BIO12,. (Temp., precipitation) etc. They are not relevant for breeding. However, the finding from Fig 7 is useful as it can inform which bioclimatic variables should be included as covariates in the GWAS model. I strongly suggest the authors re-run the GWAS model for pod drop/shattering using BIO18, BIO12, etc. as covariates. This applies to other traits such as AcoRes, ptht, Bushshape, WpWp with a strong correlation with BIOx based on Fig7. The result can then be compared with the previous GWAS. I believe this approach is better to identify the ‘true’ underlying genetic of traits.
Response 1: We thank the reviewer for the comment. We have a common garden, and the effects of environments are only coming via population structure, and population structure is already accounted for in GWAS, so this double-accounting would result in strong drops of power. Previously, we performed the study in which we were looking for associations between SNPs and bioclimatic covariates at collection sites. We added the phrase in lines 355-359 to compare results with GWAS hits from current paper: “Previously, we [25] published a study in which we were looking for associations between SNPs and bioclimatic covariates at collection sites. Two covariates, which include temperature characteristics, were jointly associated with one SNP on chromosome 8 (Ca8: 10314452). This SNP is associated with two phenotypic variables: biological yield (Byld) and plant weight without pods (WpWp) in current study”. We added paper “25. Sokolkova AB, Chang PL, Carrasquila-Garcia N, Noujdina NV, Cook DR, Nuzhdin SV, Samsonova MG (2020) The signatures of ecological adaptation in the genomes of chickpea landraces. Biophysics 65(2):237-240. doi: 10.1134/S0006350920020244” to the References section.
Point 2: I am very surprised that the authors found so many SNP associated with flower color while so few for yield (Figure S3, S4) which contradicts the fact that yield is a complex trait while flower color is a simple Mendelian trait controlled by a few genes (bHLH transcription factor.., Penmetsa RV, 2016 New Phytol) from the phenylpropanoid pathway. It seems that the authors did not point out if any known flower color genes closed to the GWAS hits. If this is not the case, what could be the reason for that?
Response 2: We thank the reviewer for the comment. Our best guess is that yield related SNPs might have smaller average effects in comparison with the color related SNPs and – accordingly – be less frequently called significant. We added the phrase “SNP Ca4: 2145082 associated with flower color (FloCol) is located upstream of Ca_07836 gene which is homologue of genes in Pisum sativum (protein A) and Medicago truncatula (bHLH-A) which are flower color associated genes [15]” in lines 277-279. We added paper “15. Hellens RP, Moreau C, Lin-Wang K, Schwinn KE, Thomson SJ, Fiers MWEJ, et al. (2010) Identification of Mendel's White Flower Character. PLoS ONE 5(10): e13230. https://doi.org/10.1371/journal.pone.0013230” to the References section.
Point 3: In the Discussion, the authors dedicated most of the content into the functions of the candidate genes which I think is important. However, I suggest the authors should consider having a small paragraph in the Discussion warning the readers to be cautious about the identified candidate genes and need further validation and functional confirmation due to, no limited to, such as long extend of LD in this germplasm, one-year observation of some complex traits like yield, seed number…
Response 3: We thank the reviewer for the comment and added the phrase “However, the identified candidate genes are needed in further validation and functional confirmation due to such factors as one-year observation of phenotypes and long extend of LD in the germplasm” to the lines 336-338.
Point 4: Minor suggestions:
L84, 55% of what? The whole Vavilov collection?
Response 4: We thank the reviewer for the comment and changed the phrase in line 81 to “This set of accessions is enriched for genotypes under cultivation a minimum of one century ago in Turkey, India, Ethiopia, Uzbekistan, and Morocco, representing the major centers of post-domestication chickpea diversification and comprising 55% of 407 analyzed accessions”.
Point 5: L98 what do you refer to with the first, third and fifth groups
Response 5: We thank the reviewer for the comment. Groups of correlated bioclimatic variables are in Supplementary Table S3. We have changed the phrases in lines 99-105 to “The first, third and fifth groups (Table S3) correspond to temperature traits. The second and fourth groups (Table S3) consist of precipitation variables. While the first group (Table S3) consists of traits with moderate positive correlation (pairwise Spearman correlation coefficient, r > 0.4, Figure 1b), traits in the second group (Table S3) have stronger positive correlations (pairwise Spearman correlation coefficient, r > 0.7, Figure 1b), and traits in the remaining groups (Table S3) have the strongest positive correlations (pairwise Spearman correlation coefficient, r > 0.9, Figure 1b)”.
Point 6: L129 Good to briefly state the reason of elevated SNP in Ca4
Response 6: We thank the reviewer for the comment and moved the phrase “We hypothesized that selection and introgression via inadvertent hybridization between more and less advanced morphotypes might have resulted in agricultural improvement genes being aggregated to genomic ‘agro islands’, and in genotype-to-phenotype relationships resembling widespread pleiotropy” from lines 420-423 to lines 132-135.
Point 7: L134 What is unlinked r2? Figure 3b is a pool of SNP from the eight chromosomes. It would be good to have a similar figure separate for the eight chromosomes in the Supplementary. This can give a clear picture of LD decay in different chromosomes. Based on the red line, LD decay at around 4Mb which is much longer than other chickpea studies with elite lines (Li et al., 2018 Front.Plnt Sci 9:190). I assume the opposite is true as landrace is supposed to be more diverse and under less selection compared to elite lines. The long extend LD in this MS reduce mapping resolution and give lower confidence for the identified candidate genes.
Response 7: We thank the reviewer for the comment. Unlinked r2 is the squared correlation coefficient (r2) between genotypes from different chromosomes. We have added LD plots separate for the eight chromosomes in the Supplementary (Figure S1) and renumbered other Supplementary figures. One suggestion to explain this pattern could be smaller genetic base in regional lines with substantial contribution of new mutations – but this remains just a speculation at present and needs more work.
Point 8: L146 I am not convinced that these characteristic warrant enough power unless the authors do an appropriate statistical power
Response 8: We thank the reviewer for this note. We replaced with “Our sample size is comparable with other recent GWAS crop publications, hopefully resulting in adequate power”.
Point 9: L186 Figure 6b The authors pooled all manhattan plots from different traits into a single plot. This is not very informative. It seems the CMplot package used by the authors can distinguish the traits using different colour or shape.
Response 9: We thank the reviewer for the comment and changed Figure 6b by plotting Manhattan plot in which we visualized significant SNPs separately for each phenotype.
Point 10: L186 Figure 6d What does it mean when you find a SNP significantly associated with a bioclimatic variable, let say, precipitation. Remember this variable is a pure environmental condition that doesn’t involve any chickpea or other plants. There is not genetics/physiology involved and is purely statistical association. I think those SNP associations are useless for crop improvement. The bioclimatic variables are actually confounding effect in GWAS and should be included as covariates instead.
Response 10: We thank the reviewer for the comment and as we mentioned earlier we have a common garden, and the effects of environments are only coming via population structure, and population structure is already accounted for in GWAS, so this double-accounting would result in strong drops of power. We changed Figure 6d by plotting Manhattan plot in which we visualized significant SNPs separately for each bioclimatic variable and renamed this plot to Figure 7. Bioclimatic variables are measured at collection sites, not at places of accessions’ phenotyping, thus identification of SNPs significantly associated with bioclimatic variables may find the signatures of historical selection for adaptation to various environmental conditions.
Point 11: L192 The bioclimatic variables are not phenotypic traits, they have no heritability!
Response 11: We thank the reviewer for the comment and deleted Figure 6c and edited Table S6.
Point 12: L198 The first eight PC explains only about 40% of the total variation (Fig S1). This again points to the author should include bioclimatic variables as covariates as described in major suggestion.
Response 12: We thank the reviewer for the comment and as we mentioned earlier we have a common garden, and the effects of environments are only coming via population structure, and population structure is already accounted for in GWAS, so this double-accounting would result in strong drops of power.
Point 13: L209 How many replicate in measuring yield and seed number? Be cautious about interpretation based on just one year, one environment.
Response 13: We thank the reviewer for the comment. There were 5 replicates in measuring yield and seed number. The description is in Supplementary Text S1: “At the time of the harvesting, the full architectural analyses had been executed for 5 randomly chosen plants per genotype”.
Point 14: L236-242 I don’t think this gene has anything to do with flowering and maturation
Response 14: We agree with the reviewer and deleted the phrases: “Another interesting gene is Ca_10428 (Ca6:2904769….2905917) that encodes a protein with a Gnk2-homologous domain. Ginkbilobin-2 (Gnk2) is an antifungal protein found in the endosperm of Ginkgo seeds, which inhibits the growth of phytopathogenic fungi such as Fusarium oxysporum. Gnk2 has considerable homology (~85%) to embryo-abundant proteins (EAP) from the gymnosperms Picea abies and P. glauca. Plant EAP are expressed in the late stage of seed maturation and are involved in protection against environmental stresses such as drought and salinity [11]”. Also, we deleted from References paper “11. Miyakawa T, Hatano K, Miyauchi Y, Suwa Y, Sawano Y, Tanokura M (2014) A secreted protein with plant-specific cysteine-rich motif functions as a mannose-binding lectin that exhibits antifungal activity. Plant Physiol. 166(2):766-78”.
Point 15: L267 The physical position of the Haploblocks in Table S9 should be indicated.
Response 15: We thank the reviewer for the comment. The positions of the haploblocks in Table S9 are in Table S8. We added the phrase “* - Haploblocks’ coordinates are in Table S8” to the Table S9.
Point 16: L284 There are many QTL detected for Ascochyta blight in chickpea in the literature (a major one in Ca4). Are there any Ascochyta Haploblocks fall in these QTL? There is also no information in term of measuring Ascochyta, how the inoculation was done, what pathotype of the pathogen, ..etc
Response 16: We thank the reviewer for the comment. Haploblocks enriched for SNPs associated with Ascochyta resistance don’t intersect with QTL detected for Ascochyta blight in the literature. The description of process of the field preparing is in Supplementary Text S1: “In the autumn of 2015, after harvesting a preceding crop of winter wheat, a continuous disking of the soil was carried out. In October the soil was plowed to a depth of 25 cm. In the spring of 2016 three-fold cultivation of the field was carried out. Prior to sowing, on April 22 the section for chickpeas was treated with "Stompe" herbicide after laying out the planting design, followed by embedding the herbicide by continuous cultivation”. Inoculation wasn’t performed. The degree of damage of Ascochyta was measured as a score and resistance to Ascochyta was measured as opposite value of the degree of damage of Ascochyta.
Point 17: L442 Is the result reported?
Response 17: We thank the reviewer for the comment. Yes, the result is in Supplementary Table S5. It is mentioned in line 177: “Central Asian and Turkish accessions are broadly distributed throughout the tree, but notably absent from groups predominated by India and Ethiopia, consistent with more extensive diversity (Table S5) at the Turkish center of origin for the species, and with longstanding but distinct secondary diversification in India, Central Asia and Ethiopia”.